# A fusion protein comprising pneumococcal surface protein A and a pneumolysin derivate confers protection in a murine model of pneumococcal pneumonia

Tanila Wood dos Santos[1,2], Pedro Almeida Gonçalves[1], Dunia Rodriguez[3], José Aires Pereira[4], Carlos Augusto Real Martinez[4], Luciana C. C. Leite[3], Lucio F. C. Ferraz[1], Thiago Rojas Converso[1], Michelle Darrieux[1]*

1 Laboratório de Microbiologia Molecular e Clínica, Universidade São Francisco, Bragança Paulista, Brazil,
2 Programa de Pós-Graduação Interunidades em Biotecnologia-USP-IPT-IB, São Paulo, Brazil,
3 Laboratório de Desenvolvimento de Vacinas, Instituto Butantan, São Paulo, Brazil, 4 Laboratório de Investigações Médicas, Universidade São Francisco, Bragança Paulista, Brazil

* sampaiomichelle@uol.com.br

## Abstract

PspA and pneumolysin are two important vaccine candidates, able to elicit protection in different models of pneumococcal infection. The high immunogenic potential of PspA, combined with a possible adjuvant effect of pneumolysin derivatives (due to their ability to interact with TLR-4) could greatly improve the immunogenicity and coverage of a protein-based pneumococcal vaccine. A chimeric protein including the N-terminal region of PspA in fusion with the pneumolysin derivative, PID1, has been shown to induce high antibody levels against each protein, and protect mice against invasive challenge. The aim of the present study was to investigate the cellular response induced by such vaccine, and to evaluate protection in a murine model of lobar pneumococcal pneumonia. Pneumococcal pneumonia was induced in BALB/c mice by nasal instillation of a high dose of a serotype 14 strain with low virulence. Airway inflammation was confirmed by total and differential cell counts in BAL and by histological analysis of the lungs, and bacterial loads were measured 7 days after challenge. Cytokine levels were determined in the bronchoalveolar fluid (BALF) of mice immunized with rPspA-PID1 fusion after challenge, by flow cytometry and ELISA. After challenge, the mice developed lung inflammation with no invasion of other sites, as demonstrated by histological analysis. We detected significant production of TNF-α and IL-6 in the BALF, which correlated with protection against pneumonia in the group immunized with rPspA-PID1. In conclusion, we found that the rPspA-PID1fusion is protective against pneumococcal pneumonia in mice, and protection is correlated with an early and controlled local inflammatory response. These results are in agreement with previous data demonstrating the efficacy of the fusion protein against pneumococcal sepsis and reinforce the potential of the rPspA-PID1 protein chimera as a promising vaccine strategy to prevent pneumococcal disease.

**Data Availability Statement:** All relevant data are within the paper and its Supporting Information files.

**Funding:** MD and LCCL received funding from Fundação de Amparo à Pesquisa do Estado de São Paulo (grants 2014/01115-9 and 2017/24832-6). TWS received a sholarship from Coordenação de aperfeiçoamento de pessoal de nível superior (CAPES).

**Competing interests:** The authors have declared that no competing interests exist.

# Introduction

*Streptococcus pneumoniae* is a leading cause of morbidity and mortality affecting mainly children and the elderly. It is estimated that 10.6 million children under five years of age are affected by pneumococcal disease annually, with more than 1.1 million deaths [1].

Strategies used to prevent pneumococcal infections are based on polysaccharides alone or in conjugation with carrier proteins [2]; however, given the limited number of serotypes in conjugate vaccines a rapid increase in pneumococcal disease caused by serotypes not included in the vaccines has been observed in many countries following implementation of pneumococcal vaccine programs [3]. Additionally, the conjugation process is complex and costly, limiting vaccine implementation in countries where disease burden is highest [2].

Therefore, serotype-independent vaccines, including whole cell and protein-based formulations, have been evaluated against *S. pneumoniae* infections. [4]. Among several proteins currently investigated [5–9] are Pneumococcal surface protein A (PspA) and Pneumolysin. PspA is an exposed protein that limits Complement deposition on the bacterium through diverse mechanisms [10] and has been shown to prevent pneumococcal killing by antimicrobial peptides [11, 12]. Pneumolysin is a cholesterol-dependent cytolysin with well documented adjuvant properties, due to its ability to interact with TLR-4 [13, 14].

Pneumolysin detoxified derivatives known as pneumolysoids (PLD) and PspA have been suggested in several studies as potent candidates for inclusion in subunit vaccines against pneumococcal infection [15–19]. The immunogenic potential of these proteins has been well characterized; however, the complexity of pneumococcal infections suggests that more than one protein must be included in the vaccine to achieve high efficacy and coverage [11].

The coadministration of PspA and PLD has shown improved protective responses when compared to each protein alone, in models of bacteremia and pneumonia [8, 20]. Another strategy to combine multiple antigens in a single formulation is the fusion of protective proteins or protein fragments, creating chimeras. Fusion proteins have been successfully evaluated in several models of pneumococcal diseases, including a rPspA-PlD1 chimera produced by our group, which was protective against pneumococcal sepsis [17]. In the present study, we evaluated the protective efficacy of the rPspA-PlD1 fusion protein against pneumococcal pneumonia, using a mouse model of focal pneumonia–which mimics the clinical features of lung colonization by pneumococci and reflects the natural course of human infections.

# Materials and methods

## Pneumococcal strains and growth conditions

The pneumococcal strains used in this study were the clinical isolates P854 (serotype 19F) and St 245/00 (Serotype 14) [21], two serotypes with low invasiveness. Pneumococci were maintained as frozen stocks (-80˚C) in Todd-Hewitt broth supplemented with 0.5% yeast extract (THY), with 10% glycerol. In each experiment, the bacteria were plated on blood agar prior to growth in THY.

## Construction of the rPspA-PlD1 chimera

The gene fragment encoding the N-terminal region of PspA 245/00 [21] was fused to the mutant detoxified Pneumolysin gene *plD1* ($Pd_{H367}$) by ligation through complementary cohesive ends added to the primers and cloned into linearized pAE-6xHis expression vector [22]. The PlD1 mutant was first described by Berry *et al.*, 1995 [23]), and retains 0.02% of the hemolytic activity of the native protein [17, 23]. The final construct, r*pspA-plD1*, was expressed in *E. coli* BL21DE3 by induction of mid-log-phase cultures with 1 mM IPTG (Sigma) and purified

through affinity chromatography with $Ni^{2+}$ charged chelating Sepharose resin (HisTrap Chelating HP; GE HealthCare) in an Akta Prime apparatus (GE HealthCare), as described by Goulart *et al.*, [17].

## Mouse pneumonia model

All animal experiments were approved by the Ethics Committee at Universidade São Francisco, Bragança Paulista–SP (CEUAUSF), (Permit Number: 001.08.12). Female BALB/c mice from Faculdade de Medicina–Universidade de São Paulo (São Paulo, Brazil) were anesthetized with 200 μL of a mixture of 0.5% xylazine and 0.25% ketamine and innoculated intranasally with $5 \times 10^6$ CFU of *Streptococcus pneumoniae* 245/00 or P854 diluted in PBS (final volume = 50 μL/animal). After 5 or 7 days, the mice (which did not show any signs of disease) were euthanized by a lethal dose of anesthetic and had their lungs collected, macerated in 1 mL of iced PBS and centrifuged at 500 x g for 10 minutes. Serial dilutions of lung homogenates were plated on blood agar and incubated overnight at 37°C for determination of bacterial counts by CFU (Colony forming units). Comparison between 5 and 7 days of infection with each pneumococcal strain was performed using Student t test. Cellular influx in the lungs was evaluated by differential leucocyte count in BALF samples collected after the mice were euthanized at different time periods following challenge (0, 3, 6, 12, 24 and 48 h).

## Immunization and challenge

Female BALB/c mice were immunized subcutaneously with 3 doses of 8.8 μg of rPspA1, 11.2 μg of PlD1, a mixture of the proteins (rPspA1+PlD1, co-administered) or 20 μg of the hybrid rPspA1-PlD1 at 14-day intervals, using sterile saline solution 0.9% with 50 μg of Al$(OH)_3$ as adjuvant (in a final volume of 100 μL). The adjuvant alone in saline was used as a control. Two weeks after the last immunization, the mice were anesthetized and challenged by intranasal inoculation with 50 μL of PBS containing $5 \times 10^6$ CFU of St 245/00. After different time points, the animals were euthanized and the BALF was collected for determination of cellular infiltrate and cytokine production. The lungs were removed for determination of bacterial loads and histological analysis of the inflammatory infiltrate by H.E. staining. Blood and liver were also collected for CFU count. Differences between pneumococcal counts in each group were analyzed by Mann–Whitney U test.

## Analysis of cytokine production in BALF

Cytokine production was evaluated in the BALF of immunized mice after 6, 12, 24 and 48 hours of challenge using Cytometric Bead Array (CBA, BD Biosciences). The following cytokines were analyzed: IFN-γ, TNF-α, IL-2, IL-4, IL-6, IL-10 and IL-17a. Twenty-five microliter aliquots of BALF of each mouse were incubated in presence of antibody-conjugated beads and the fluorescence analyzed using FACS CANTO (BD Biosciences), according to the manufacturer's instructions. Production of IL-6 and TNF-α was confirmed by ELISA (P-Protech).

# Results

## Murine model of pneumococcal pneumonia

After 5 days of intranasal inoculation with the pneumococcal strains St 245/00 and P854, the mice lungs were colonized with pneumococci, and the bacterial levels increased after seven days (Fig 1A). Although both bacteria behaved similarly in the challenge experiment, the bacterial counts were a little less disperse with St 245/00, as indicated by the slightly lower standard deviation bar on this group. Based on those results, we have proceeded with the challenge

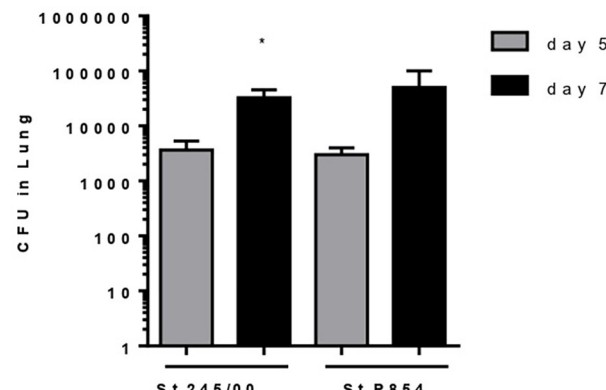

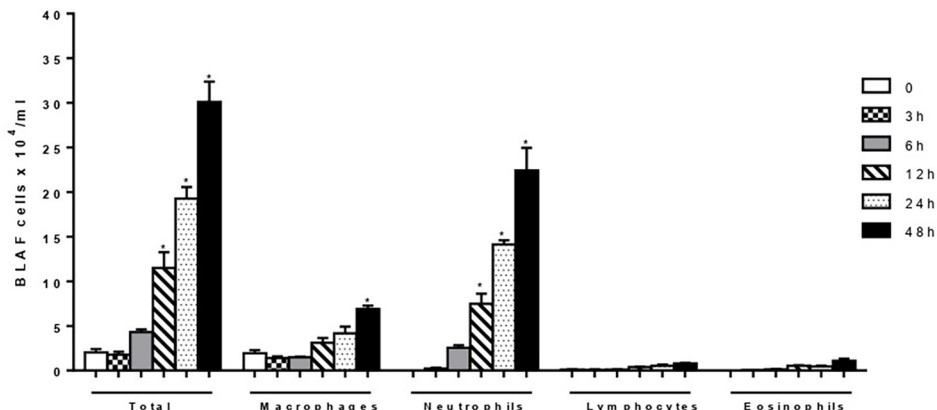

**Fig 1. Mouse model of pneumococcal pneumonia.** (A) Bacterial counts were determined in the lungs of nonimmunized mice on days 5 and 7 after intranasal inoculation with pneumococcal strains St 245/00 (serotype 14) and P854 (serotype 19F). Values were compared for each strain after 5- and 7 days using Student t test (*p<0,05). (B). Leucocyte infiltrates in the BALF were calculated for different time points after intranasal challenge with pneumococcal strain St 245/00. Statistical analysis was performed using one way ANOVA with Dunnet's posttest. *p<0,05 in comparison with cells counts at 0 hours.

experiments using St 245/00 and 7 days as the colonization endpoint. Analysis of the cellular infiltrates revealed a significant influx of leucocytes in the BALF samples after 12 h of infection with ST 245/00, which continued to increase until 48 h (Fig 1B). The infiltrate was mainly composed of neutrophils, followed by macrophages.

## Effect of immunization on lung colonization by *Streptococcus pneumoniae*

Aiming to evaluate the effect of immunizations over infection, lungs, liver, and blood were collected from immunized mice at various time points after intranasal challenge with St 245/00 and CFU evaluated. Bacterial loads in the lungs of mice immunized with the fusion protein, rPspA-PlD1, remained very low throughout the experiment (Fig 2A). In the group immunized with rPspA alone, there was an increase in bacterial counts after 2 h of infection, followed by a gradual decrease over the next hours. The control and PlD1 groups showed some fluctuation

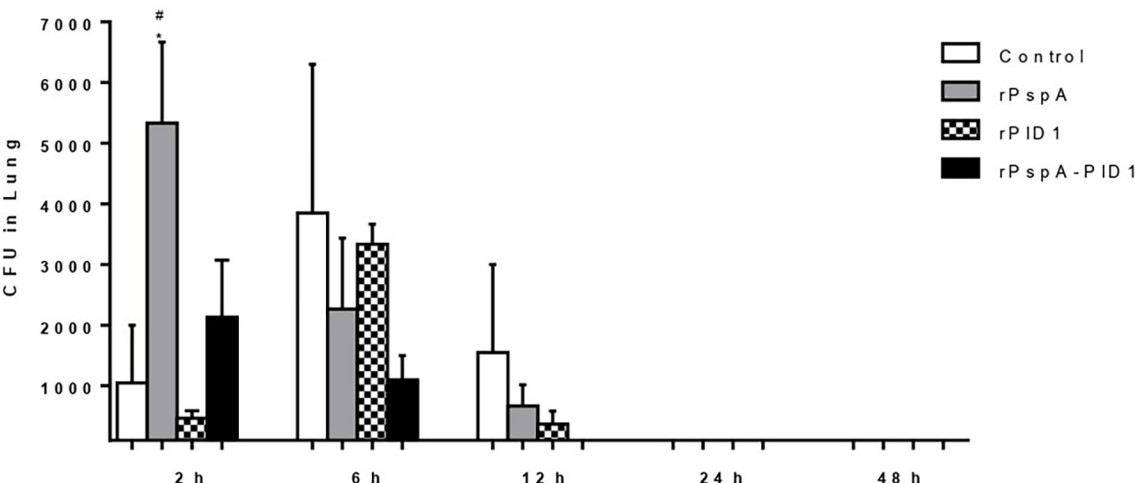

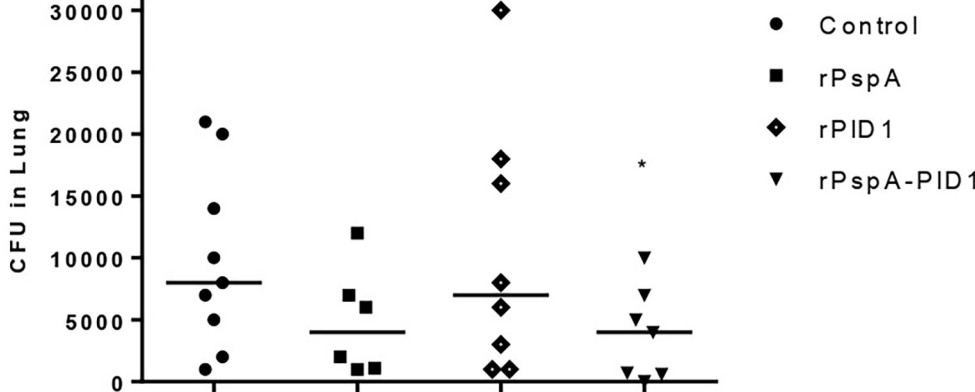

**Fig 2. Lung colonization by *Streptococcus pneumoniae* in mice immunized with rPspA_PlD1.** Mice immunized with 3 doses of rPspA, rPlD1, or the fusion protein in Alum were challenged intranasally with $5 \times 10^6$ CFU of St 245/00. Bacterial counts in the lungs of immunized and control mice (injected with Alum diluted in PBS) are shown after 2–48 h (A) or seven days (B). (*p<0.05 for the same immunization group at different time points and #p<0.05 for immunized x control mice).

in the number of pneumococci over time, but there were no significant differences between these groups. After 7 days of challenge, however, the animals immunized with the fusion protein had significantly lower bacterial loads in the lungs in comparison with the control group (Fig 2B), and 1 out of 7 mice was cleared from bacteria. The group immunized with PspA alone displayed a tendency towards lower CFU counts in the lungs, but it did not reach statistical significance. The group that received PlD1 alone had bacterial loads comparable to the sham-immunized control. No bacteria could be detected in the blood and liver of the immunized or control mice at any time point (data not shown).

## Characterization of the cellular immune responses induced by immunization and challenge

The cellular infiltrates in the BALF of immunized and challenged mice were evaluated at different time points after challenge. The immunized groups show an early cellular influx in the lungs after pneumococcal challenge, in comparison with the control group, which peaked at 48 h (Fig 3A). The lungs of mice immunized with the rPspA-PlD1 fusion protein exhibited a rapid increase in immune cells after 6 hours of infection, which peaked at 12 h and was followed by a marked reduction at 24 h. Mice immunized with rPspA alone did not present a significant cellular infiltrate at any time point, while the rPlD1 group showed an increase in BALF leucocytes at 6 h post-infection, which persisted until 24 h. Histological analysis of the lungs confirmed the inflammatory infiltrates found in BALF (Fig 3B and 3C). The lungs of immunized and challenged mice presented an early low inflammatory response, with a discrete leucocyte influx and alveolar integrity (Fig 3B). In contrast, the control group showed a moderate and delayed inflammatory response, with congestion, alveolar cell hyperplasia and increased levels of leucocyte infiltrates (Fig 3C).

Cytokine production in the BALF was determined by CBA kit (S1 Appendix) and confirmed by ELISA. Analysis of TNF-α production revealed an early increase in the groups immunized with rPlD1 and the hybrid protein and a delayed response in the control group, which peaked at 12 hours (Fig 4A). At 24 h, the levels of TNF-α decreased significantly in all groups tested. The immunized and control groups showed an increase in IL-6 production 12 h after challenge; however, the levels of IL-6 were significantly higher in the BALF of mice immunized with the hybrid protein in comparison with the other groups (Fig 4B). IL-6 production remained high in this group after 24 h, with a marked reduction after 7 days. The results confirmed previous flow cytometry data obtained using CBA kit, which showed an early increase in TNF-α in the immunized groups and significantly higher levels of IL-6 in the group immunized with the hybrid within the first six hours of infection.

## Discussion

Protein-based vaccines are an interesting alternative to prevent pneumococcal infections; they can provide serotype-independent protection, overcoming the serotype-replacement observed with the current conjugate vaccines. PspA and Pneumolysin are important virulence factors which have been extensively studied as pneumococcal vaccine candidates, with encouraging results in several infection models as well as clinical trials [4].

The main limitation of pneumococcal protein-based vaccines has been the low immunogenicity of isolated proteins and, in the case of PspA, the high structural and serological variability [24], which hamper the protective efficacy of these proteins. An alternative to surpass these limitations is to combine different proteins in a single formulation. Protein chimeras–where the most protective fragments of each protein are expressed in fusion forming a new molecule–are an interesting approach, since they can be produced as a single antigen, combining the protective effects of multiple antigens. Previous studies using protein chimeras have suggested protective responses against all the antigens included in the formulations [15, 17, 25, 26]. We have previously constructed a fusion protein including the N-terminal region of a family 1 PspA (which was selected based on its ability to induce antibodies displaying high cross-reactivity with heterologous rPspA molecules–[21] and a detoxified pneumolysin derivative with a His367-Arg substitution [17, 27]. The final construct, rPspA-PlD1, was initially investigated as a vaccine candidate against pneumococcal sepsis. Immunization with the fusion protein induced high levels of opsonic antibodies and protection against pneumococcal sepsis, which correlated with increased antibody-mediated complement deposition on the

A

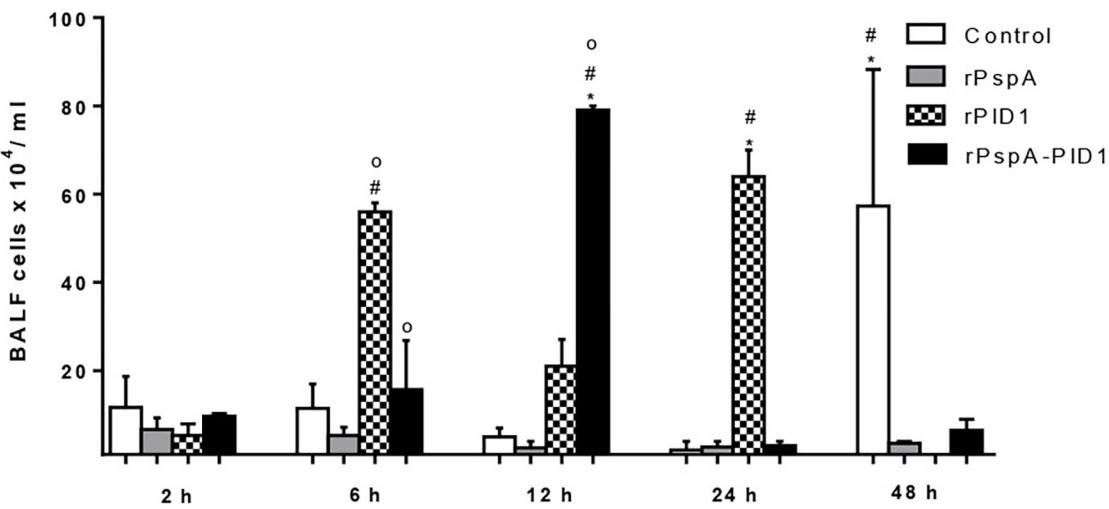

B

C

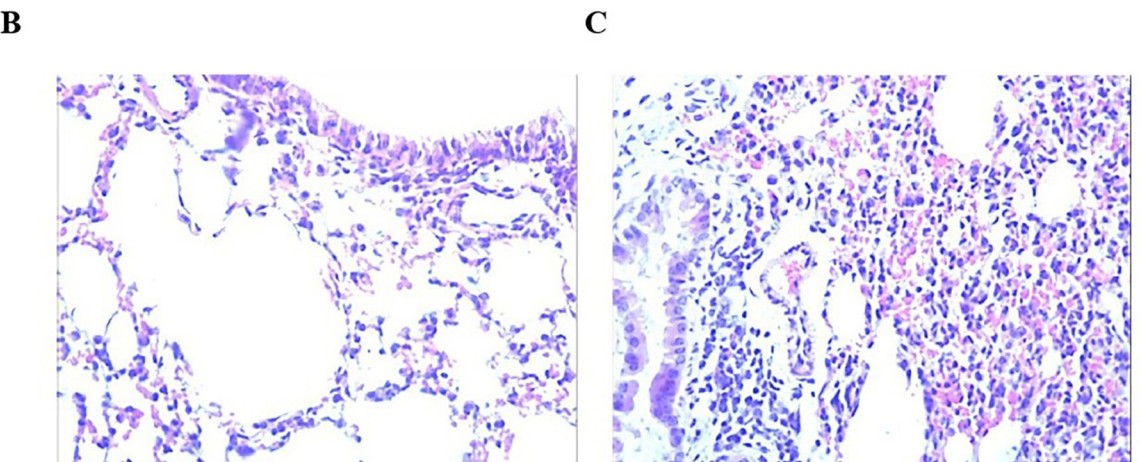

**Fig 3. Induced immune responses in lungs and BALF.** (A) Cellular infiltrate in the BALF following pneumococcal challenge. Immunized and control mice were euthanized at different time points after intranasal inoculation of St 245/00 and the total cell counts in the BALF were determined and compared among immunization groups and times points. Statistical analysis was performed using ANOVA with Tukey's and Dunnet's posttests. (*) $p<0,05$ between samples from the same immunization group at different times after challenge; (#) $p<0,05$ in comparison with the control (Alum) at the same time point; (o) $p<0,05$ when comparing immunization with rPspA-PlD1 versus isolated proteins at the same time point. (B and C) Histological analysis of lung tissue after pneumococcal challenge. 1 μm sections of the left lung lobe from mice immunized with the hybrid (B) and from control mice (C) after 48 h were stained with hematoxylin-eosin, showing slight and moderate levels of inflammation, respectively (Original magnification, x400).

bacterial surface [17]. However, the most common outcome of pneumococcal invasive infections is not sepsis, but lobar pneumonia. Therefore, in the present study, the rPspA-PlD1 hybrid was investigated as a vaccine candidate in a mouse model of pneumonia, which better reflects the hallmarks of pneumococcal infection in the human host.

Initially, a model of pneumococcal lobar pneumonia was developed by intranasal inoculation with a pneumococcal strain displaying low virulence in mice, as proposed by Briles and

A

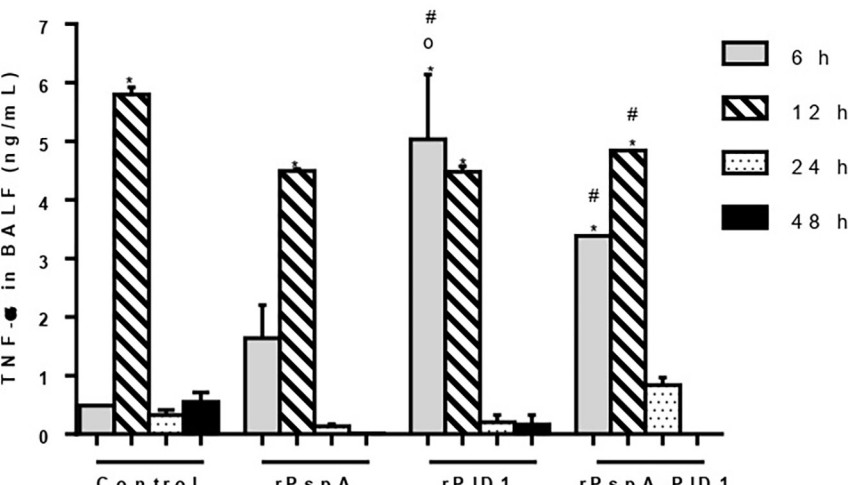

B

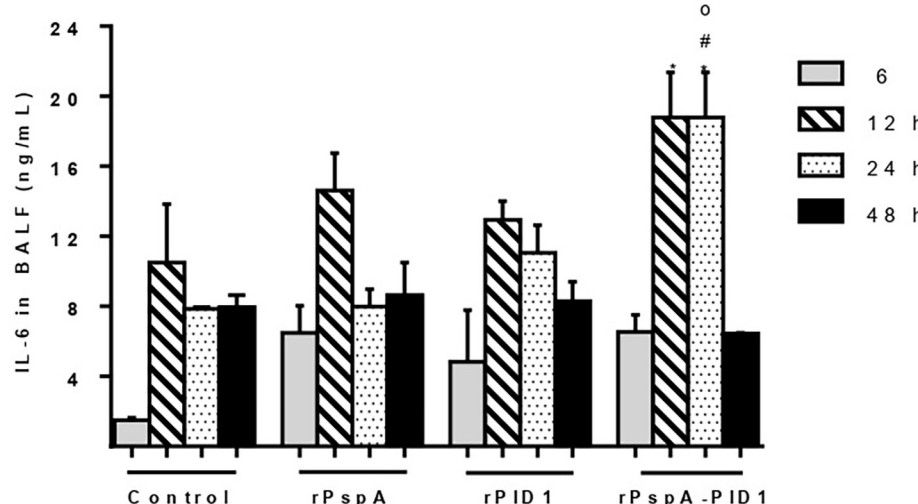

**Fig 4. Cytokine production by immunized mice after pneumococcal challenge.** Production of TNF-α (A) and IL-6 (B) in the BALF of immunized mice was detected by ELISA and compared with control mice (receiving adjuvant in PBS). *$p<0.05$ for the same group at different time periods; #$p<0.05$ as compared with the control group;°$p<0.05$ for the rPspA-PlD1 group as compared to the individual proteins.

cols [20]. In the present study, we have tested two strains of different serotypes, 14 and 19F, which express PspAs belonging to family 1 (clades 1 and 2, respectively). Challenge with the serotype 14 strain, St 245/00, resulted in lung colonization at day five, which increased at day seven, with less variations among individuals. Therefore, this strain was chosen to evaluate the effects of vaccination on pneumonia.

Next, the pneumonia model was used to investigate protection induced by subcutaneous immunization with the recombinant proteins. Only the group immunized with the fusion protein showed a significant reduction in pneumococcal colonization after 7 days of infection; although the PspA group showed a tendency towards lower bacterial counts in the lungs, none of the individual proteins were protective in this model. This result reinforces the enhanced effectiveness of chimeric proteins against pneumococcal lung infection. A similar pneumonia model has been previously used to evaluate the protective efficacy of rPspA and a pneumolysoid, rPdB. In that study, the combination of rPspA and rPdB promoted the strongest inhibition in lung colonization, confirming the importance of including different antigens to increase vaccine efficacy [20]. The present work supports those findings, and further demonstrates that fusion proteins maintain the protective efficacy of combined antigens, with the advantage of being produced as a single molecule. This greatly impacts the costs associated with vaccine production and could, therefore, contribute to a wider distribution of the vaccine among developing countries, in which the burden of pneumococcal diseases is higher.

Protection elicited by the vaccine was associated with early, controlled inflammatory responses, represented by cellular infiltrates and an inflammatory cytokine profile. Immunized mice showed an early, yet discrete, increase in IL-6 and TNF-α in BAL fluids, while the control group had a delayed response upon infection. A similar effect has been observed previously in mice immunized with protein-based pneumococcal vaccines, and was, as in the present work, correlated with protection against systemic infection [28–31].

A study with IL-6 deficient mice has shown that these animals had higher pneumococcal loads in the lungs after 40 hours and died earlier than the wild-type group [31]. Also, the quick production of TNF-α in the group vaccinated with fusion protein, in the first hours of infection has been predictive of better infection outcomes, including less tissue damage and better survival in mice [28–30].

Our group has previously shown that the rPspA-PlD1 construct was able to protect mice against sepsis through the induction of protective antibodies [17]; here we demonstrated that immunization with this protein also elicited cytokine production related to protection in the pneumonia model. This data is supported by Wilson *et al.*, who have demonstrated that protection against lung infections requires humoral and cell-mediated immune responses [32].

In conclusion, we have demonstrated that the fusion protein rPspA-PlD1, including the protection eliciting N-terminal fragment of PspA and a detoxified pneumolysoid, is a strong candidate for future serotype-independent pneumococcal vaccines, able to promote protective responses against systemic infection as well as lobar pneumonia. Protection has been associated with production of opsonic antibodies and complement-mediated phagocytosis [17] and controlled inflammatory responses in the BALF of subcutaneously immunized mice.

## Supporting information

**S1 Appendix. Cytokine production analysis by CBA kit.** The cytokine production was analyzed in the BALF after 6, 12, 24 and 168 hours after infection using the Cytometric Bead Array (CBA, BD Biosciences).
(DOCX)

## Author Contributions

**Conceptualization:** Thiago Rojas Converso, Michelle Darrieux.

**Data curation:** Tanila Wood dos Santos, Dunia Rodriguez, José Aires Pereira, Carlos Augusto Real Martinez.

**Formal analysis:** Luciana C. C. Leite, Michelle Darrieux.

**Funding acquisition:** Luciana C. C. Leite, Michelle Darrieux.

**Investigation:** Tanila Wood dos Santos, Pedro Almeida Gonçalves, Dunia Rodriguez, Carlos Augusto Real Martinez.

**Methodology:** Tanila Wood dos Santos, Pedro Almeida Gonçalves, Dunia Rodriguez, José Aires Pereira, Thiago Rojas Converso.

**Project administration:** Michelle Darrieux.

**Supervision:** Michelle Darrieux.

**Validation:** Lucio F. C. Ferraz.

**Writing – original draft:** Tanila Wood dos Santos, Pedro Almeida Gonçalves, Michelle Darrieux.

**Writing – review & editing:** Dunia Rodriguez, Carlos Augusto Real Martinez, Luciana C. C. Leite, Lucio F. C. Ferraz, Thiago Rojas Converso, Michelle Darrieux.

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
