## [Decision Letter · Decision Letter 0]

21 Sep 2022

PONE-D-22-17627A fusion protein comprising Pneumococcal surface protein A and a Pneumolysin derivate confers protection in a

murine model of pneumococcal pneumonia.PLOS ONE

Dear Dr. Darrieux,

Thank you for submitting your manuscript to PLOS ONE. After careful consideration, we feel that it has merit but does not fully meet PLOS ONE’s publication criteria as it currently stands. Therefore, we invite you to submit a revised version of the manuscript that addresses the points raised below during the review process.

We look forward to receiving your revised manuscript.

Kind regards,

Ray Borrow, Ph.D., FRCPath

Academic Editor

PLOS ONE

Journal Requirements:

2. As part of your revision, please complete and submit a copy of the Full ARRIVE 2.0 Guidelines checklist, a document that aims to improve experimental reporting and reproducibility of animal studies for purposes of post-publication data analysis and reproducibility: https://arriveguidelines.org/sites/arrive/files/Author%20Checklist%20-%20Full.pdf (PDF). Please include your completed checklist as a Supporting Information file. Note that if your paper is accepted for publication, this checklist will be published as part of your article

Reviewers' comments:

Reviewer's Responses to Questions

**Comments to the Author**

1. Is the manuscript technically sound, and do the data support the conclusions?

Reviewer #1: Yes

Reviewer #2: Yes

2. Has the statistical analysis been performed appropriately and rigorously? 

Reviewer #1: Yes

Reviewer #2: Yes

3. Have the authors made all data underlying the findings in their manuscript fully available?

Reviewer #1: No

Reviewer #2: Yes

4. Is the manuscript presented in an intelligible fashion and written in standard English?

Reviewer #1: Yes

Reviewer #2: No

5. Review Comments to the Author

Reviewer #1: Overall this is a reasonable manuscript that follows up on a previous study. For the most part, the data supports the conclusions. Two points should be addressed. 1. While the purpose was to examine protection against pneumonia as opposed to sepsis, why was this data not included in the sepsis manuscript. There have already been pneumonia challenge models described for the pneumococcus. The authors could have use one of those rather than develop their own. 2. The authors state that their model had pneumococci restricted to the lungs and no other organs. However, there is no data to support this clam.

Reviewer #2: Summary observations

Abstract is clear and so is the introduction. The authors demonstrate the scientific basis for initiating the study. Appropriate references are quoted, institutional review board’s approval was obtained. The materials and methods section (mainly the Mouse pneumonia model part) need needs to be modified to reflect what is presented in the results section (Murine Model of pneumonia) (see Specific comments). The data were collected, analyzed, and interpreted by and large correctly. Figures are clear and readable but there are mistakes in some of the Figures. The results are adequately discussed.

Specific Comments

Introduction

L60 and L74 may have to be rewritten to avoid starting a new paragraph with a backward link to the previous paragraph but to start with a fresh topic sentence. Where there is a backward link, readers may conclude that this is simply ‘more of the same’ and so skip onwards to the next paragraph.

Materials and Methods

L89 and L126 ‘St’ is written in italic while elsewhere it is not (see L144, L147, L153. L199 and L246). In L177 it is ‘Sp’.

L144-147(Results section) and L254-258 - the authors present and discuss ‘lung colonization and determination of colonization end-point’; 5 days versus 7 days. However, information about comparing St 245/00 and P854 colonization of the lung (5 days versus 7 days) and determination of the endpoint is missing in the materials and methods section (mouse and pneumonia model). In fact, L111 starts with the words ‘After 7 days’ giving the impression that the comparison was done for ‘day 7’ only.

L116 - “Comparison between groups was performed using Students t test’. It is not clear which groups the authors are referring to. It is confusing. Is it BALF verses Lung? Or St 245/00 versus P854. Clarify.

Results

L114 (Methods-Mouse pneumonia model) reads ‘Serial dilutions of BALF…………were plated on blood agar…. for determination of bacterial counts …’. However, no data on BALF bacterial counts are provided; only CFU in the lung in presented (see Figure 1A)

In Figure 1A, the authors used ‘Lung CFU’ (Y-axis label) and in Figures 2A and 2B used ‘CFU in BALF’ and ‘CFU in Lung’ respectively. For consistency’s sake use the same ‘style’ of labelling.

L126 - L128. BALF was collected after seven days to determine cellular infiltrate and cytokine production. However, Figures 3A and 4A &B are for cellular infiltrate and cytokine production for time points 2-48 hrs and not for day 7. Clarify the ‘‘After seven days.’

L129 reads ‘Blood and liver were also collected for CFU count’. No data are provided.

L146 -L147 read ‘…we have proceeded with the challenge experiments using St 245/00 and 7 days as the colonization endpoint.’ and L255 goes ‘Challenge with the serotype 14 strain, St 245/00, resulted in lung colonization at day five, which increased at day seven, with less variations among individuals. Therefore, this strain was chosen to evaluate the effects of vaccination on pneumonia.’ Given that both St 245/00 and P854 gave ‘comparable’ results (Figure 1A), it would appear then that St245/00 was chosen on the basis of ‘less variations among individuals., However, the data on ‘variations among individuals’ have not been provided.

L168-L169 and L178 refer to bacterial loads in the BALF but Figure 2B is about ‘CFU in Lung”

Figure 1B-The Y axis label should be ‘BALF cells x 104/ml’ not just ‘cells x 104/ml’

Figure 4A and B. In the legend, it should be 48 hrs not 168hrs

6. PLOS authors have the option to publish the peer review history of their article (what does this mean?). If published, this will include your full peer review and any attached files.

Reviewer #1: No

Reviewer #2: **Yes: **Enoch Sepako, PhD

---

## [Author Response · Author response to Decision Letter 0]

11 Oct 2022

We would like to thank the reviewers for the careful analysis of the manuscript. 

All the suggestions were accepted and incorporated in the final version (the modifications are highlighted in yellow). We have addressed each comment individually, as follows:

Reviewer #1: Overall this is a reasonable manuscript that follows up on a previous study. For the most part, the data supports the conclusions. Two points should be addressed. 1. While the purpose was to examine protection against pneumonia as opposed to sepsis, why was this data not included in the sepsis manuscript. There have already been pneumonia challenge models described for the pneumococcus. The authors could have use one of those rather than develop their own. 2. The authors state that their model had pneumococci restricted to the lungs and no other organs. However, there is no data to support this clam.

1. The sepsis data was generated earlier as part of the project of another student at the lab. She was comparing different vaccine formulations including Pspa and pneumolysoids, and she focused on the sepsis model, which was already well stablished in our lab. The pneumonia model was stablished later, when we identified pneumococcal strains of serotypes with lower invasiveness to use in this infection model. We have, in fact, based our pneumonia model in previously described work, but we selected strains from our bacterial bank and therefore some adjustments were necessary due to particularities in these specific strains.

2. No bacteria were found in the blood and liver of the immunized or control mice in any time point after challenge. This information was added to the text (lines 170-171).

Reviewer #2:

Summary observations

Abstract is clear and so is the introduction. The authors demonstrate the scientific basis for initiating the study. Appropriate references are quoted, institutional review board’s approval was obtained. The materials and methods section (mainly the Mouse pneumonia model part) need needs to be modified to reflect what is presented in the results section (Murine Model of pneumonia) (see Specific comments). The data were collected, analyzed, and interpreted by and large correctly. Figures are clear and readable but there are mistakes in some of the Figures. The results are adequately discussed.

Specific Comments 

Introduction

L60 and L74 may have to be rewritten to avoid starting a new paragraph with a backward link to the previous paragraph but to start with a fresh topic sentence. Where there is a backward link, readers may conclude that this is simply ‘more of the same’ and so skip onwards to the next paragraph.

These sentences have been modified as suggested

Materials and Methods

L89 and L126 ‘St’ is written in italic while elsewhere it is not (see L144, L147, L153. L199 and L246). In L177 it is ‘Sp’.

The abbreviations have been corrected and kept as St throughout the manuscript.

L144-147(Results section) and L254-258 - the authors present and discuss ‘lung colonization and determination of colonization end-point’; 5 days versus 7 days. However, information about comparing St 245/00 and P854 colonization of the lung (5 days versus 7 days) and determination of the endpoint is missing in the materials and methods section (mouse and pneumonia model). In fact, L111 starts with the words ‘After 7 days’ giving the impression that the comparison was done for ‘day 7’ only.

The section was corrected as indicated.

L116 - “Comparison between groups was performed using Students t test’. It is not clear which groups the authors are referring to. It is confusing. Is it BALF verses Lung? Or St 245/00 versus P854. Clarify.

The comparison was performed between days 5 and 7 of infection with each strain. This information was added to the manuscript. 

Results

L114 (Methods-Mouse pneumonia model) reads ‘Serial dilutions of BALF…………were plated on blood agar…. for determination of bacterial counts …’. However, no data on BALF bacterial counts are provided; only CFU in the lung in presented (see Figure 1A)

The reviewer is correct. For bacterial counts, we have collected and plated the lung homogenates at days 5 and 7 post-infection. We used the BALF for analysis of the cellular infiltrates at 0, 3, 6, 12, 24 and 48 h following challenge. This information was corrected in the methods section.

In Figure 1A, the authors used ‘Lung CFU’ (Y-axis label) and in Figures 2A and 2B used ‘CFU in BALF’ and ‘CFU in Lung’ respectively. For consistency’s sake use the same ‘style’ of labelling.

The figure has been adjusted as indicated.

L126 - L128. BALF was collected after seven days to determine cellular infiltrate and cytokine production. However, Figures 3A and 4A &B are for cellular infiltrate and cytokine production for time points 2-48 hrs and not for day 7. Clarify the ‘‘After seven days.’

The information was corrected on the text. The animals were euthanized at various time points. 

L129 reads ‘Blood and liver were also collected for CFU count’. No data are provided.

No bacteria were found in the blood and liver of the immunized or control mice in any time point after challenge. This information was added to the text (lines 170-171).

L146 -L147 read ‘…we have proceeded with the challenge experiments using St 245/00 and 7 days as the colonization endpoint.’ and L255 goes ‘Challenge with the serotype 14 strain, St 245/00, resulted in lung colonization at day five, which increased at day seven, with less variations among individuals. Therefore, this strain was chosen to evaluate the effects of vaccination on pneumonia.’ Given that both St 245/00 and P854 gave ‘comparable’ results (Figure 1A), it would appear then that St245/00 was chosen on the basis of ‘less variations among individuals., However, the data on ‘variations among individuals’ have not been provided.

Although both bacteria behaved similarly in the challenge experiment, the results were a little less disperse with St 245/00, as indicated by the slightly lower standard deviation bar on this group. This information was added to the manuscript (lines 143-145).

L168-L169 and L178 refer to bacterial loads in the BALF but Figure 2B is about ‘CFU in Lung”

The information was corrected in the text.

Figure 1B-The Y axis label should be ‘BALF cells x 104/ml’ not just ‘cells x 104/ml’

Figure 1B has been adjusted as indicated.

Figure 4A and B. In the legend, it should be 48 hrs not 168hrs

The figure legends have been

---

## [Editor Report · Decision Letter 1]

25 Oct 2022

A fusion protein comprising Pneumococcal surface protein A and a Pneumolysin derivate confers protection in a

murine model of pneumococcal pneumonia.

PONE-D-22-17627R1

Dear Dr. Darrieux,

We’re pleased to inform you that your manuscript has been judged scientifically suitable for publication and will be formally accepted for publication once it meets all outstanding technical requirements.

Kind regards,

Ray Borrow, Ph.D., FRCPath

Academic Editor

PLOS ONE
---

## [Editor Report · Acceptance letter]

2 Nov 2022

PONE-D-22-17627R1 

A fusion protein comprising Pneumococcal surface protein A and a Pneumolysin derivate confers protection in a murine model of pneumococcal pneumonia 

Dear Dr. Darrieux:

I'm pleased to inform you that your manuscript has been deemed suitable for publication in PLOS ONE. Congratulations! Your manuscript is now with our production department. 

Kind regards, 

on behalf of

Prof. Ray Borrow 

Academic Editor

PLOS ONE